# Microbiota-Derived β-Amyloid-like Peptides Trigger Alzheimer’s Disease-Related Pathways in the SH-SY5Y Neural Cell Line

**DOI:** 10.3390/nu13113868

**Published:** 2021-10-29

**Authors:** Aitor Blanco-Míguez, Hector Tamés, Patricia Ruas-Madiedo, Borja Sánchez

**Affiliations:** 1Departamento de Microbiología y Bioquímica, Instituto de Productos Lácteos de Asturias (IPLA), Consejo Superior de Investigaciones Científicas (CSIC), Paseo Río Linares S/N, 33300 Villaviciosa, Asturias, Spain; aitor.blancomiguez@unitn.it (A.B.-M.); hector.tames@ipla.csic.es (H.T.); ruas-madiedo@ipla.csic.es (P.R.-M.); 2CIBIO—Dipartimento di Biologia Cellulare, Computazionale e Integrata, University of Trento, Via Sommarive 9, 38123 Povo, Italy; 3Functionality and Ecology of Beneficial Microbes (MicroHealth) Group, Instituto de Investigación Sanitaria del Principado de Asturias (ISPA), 33011 Oviedo, Asturias, Spain

**Keywords:** Alzheimer’s disease, human gut microbiota, β-amyloid-like peptides

## Abstract

Here, we present the first in silico and in vitro evidence of Aβ-like peptides released from meaningful members of the gut microbiome (mostly from the Clostridiales order). Two peptides with high homology to the human Aβ peptide domain were synthesized and tested in vitro in a neuron cell-line model. Gene expression profile analysis showed that one of them induced whole gene pathways related to AD, opening the way to translational approaches to assess whether gut microbiota-derived peptides might be implicated in the neurodegenerative processes related to AD. This exploratory work opens the path to new approaches for understanding the relationship between the gut microbiome and the triggering of potential molecular events leading to AD. As microbiota can be modified using diet, tools for precise nutritional intervention or targeted microbiota modification in animal models might help us to understand the individual roles of gut bacteria releasing Aβ-like peptides and therefore their contribution to this progressive disease.

## 1. Introduction

Alzheimer’s disease (AD) is a neurological disorder characterised by a progressive and gradual decline in cognitive function. AD is neuropathologically distinguished by the presence of neuropil threads, the loss of specific neurons and synapses, and the occurrence of senile plaques. At the molecular level, extracellular plaque deposits of the β-amyloid (Aβ) peptide and the intracellular accumulation of neurofibrillary tangles containing hyperphosphorylated Tau protein can be noticed [1]. 

The Aβ peptide is the result of a cleavage derived from the transmembrane domain of the amyloid precursor protein (APP), a receptor protein of the glycosylated, integral membrane cell surface, with 695 amino acids [2]. Most of the Aβ peptides are 40 amino acids long (Aβ40), however, peptides with lengths between 38 and 43 amino acids were found in vivo [3]. Among them, Aβ42 has been shown to be more hydrophobic and amyloidogenic [4]. The increase of Aβ42 levels seems to be related to oligomerisation, fibrillation and amyloid plaque generation [5,6].

In a previous study, we have reported on the presence of potential neuropeptides in human gut microbiota using the MAHMI database [7]. Surprisingly, few of them showed similarity to β-amyloid peptide (Aβ). Independent scientific works have shown relationships between the gut microbiome and several neurological disorders. Although few studies have been focused on Alzheimer’s Disease (AD), the ability of some bacteria to produce amyloid-like fibrils has already been demonstrated, as has the accumulation of β-amyloid (Aβ) peptide plaques in enteric neurons of mouse models of AD and in the submucosa of AD patients. For instance, independent scientific works have linked altered microbiota populations with a series of neurological disorders [8,9,10].

This scenario, and previous works showing the in vitro activity of immunomodulatory peptides [11] derived from the human gut microbiome, prompted the investigation of potential AD-promoting peptides encrypted in the human gut exoproteome. In this exploratory work we report the first in silico and in vitro evidence of β-amyloid-like peptides released from meaningful members of the gut microbiome. This work provides preliminary evidence that such peptides might be implicated in the neurodegenerative process that initiates, promotes or mediates AD, and therefore might open new research paths into the therapeutic modulation of gut microbiota in AD.

## 2. Materials and Methods

### 2.1. Data Retrieval and Structure Prediction

The sequence data of 91,325,790 potential bioactive peptides from the MAHMI database [12] were aligned against the Aβ precursor protein (UniProtKB-P0506) and the microtubule-associated protein TAU (UniProtKB-P10636) using BLASTp (e-value < 1 × 10^−5^; Appendix A).

The secondary structures of the Aβ-like peptides were predicted using the PSIPRED. The tertiary structures of the Aβ40 and Aβ42 peptides were retrieved from the PDB database (https://www.rcsb.org/structure/1aml), and secondary structure was extracted from the PDB files. 

Secondary structure alignment was performed via the alignment procedure described by Przytycka et al. [13]. The tertiary structures of the Aβ-like peptides were predicted using ExPASy SWISS-MODEL and RPBS PEP-FOLD. Structure alignment was performed using TM-Align (Appendix A).

### 2.2. Maintenance and Differentiation of the SH-SY5Y Cell Line

SH-SY5Y was purchased from the European Collection of Authenticated Cell Cultures (ECACC 94030304, Salisbury, UK) and routinely cultured and subcultured, following manufacturer instructions, in 1:1 mixture of EMEM (Eagle’s Minimum Essential Medium) and F12 medium (hereafter EMEM/F12) with the required supplements (10% *v/v* of heat-inactivated foetal serum bovine, 50 U/mL penicillin, 50 ug/mL streptomycin, 50 ug/mL gentamicin and 2 mM L-Glutamine). For neuronal differentiation, cells adhered to the flasks or wells were treated with complete EMEM/F12 supplemented with 10 uM all-trans retinoic acid for 5 days, with medium renewal on the third day. All reagents were purchased from Sigma-Merck (Darmstadt, Germany).

### 2.3. Cultivation of the Cell Lines with the DF56 and AG25 Peptides and RNA-Seq Analyses

SH-SY5Y cells were seeded into several 6-well plates (BD Falcon-Corning, Corning, NY, USA) and allowed to reach confluence (>90%), before inducing the differentiation as previously indicated. Cells were challenged over 3 h with AG25 or DF56 peptides at 1 and 2 ug/mL, and the corresponding DMSO controls (0.05% and 1%, respectively); a negative control (EMEM/F12 medium) was also included. The seven conditions were tested in triplicate. Rationale for the selection of the concentrations used in this work is explained in Appendix A. RNA was directly extracted from the cells using the RNeasy mini kit (Qiagen, GmBH, Hilden, Germany) following the manufacturer’s instructions but avoiding the use of the RNAprotect reagent. The 21 samples were delivered under dry-ice conditions to GenProbio S.R.L. (Parma, Italy), where they were sequenced (18–22 M reads/sample, 75 nt paired ends reads).

Raw FASTq reads were trimmed, filtered and mapped against the human HG38 genome. Further, transcript quantification and gene-set enrichment analysis was performed using standard analysis (Appendix A). 

## 3. Results

In this proof of concept, we present our primary and early findings of research data derived from a bigger microbial peptidome survey, which showed the existence of Aβ-like peptides encrypted in the human gut metaproteome. The 91,325,790 potentially bioactive peptides available in the MAHMI database, characterising the immunomodulatory potential of the human gut microbiome, were subjected to a large-scale screening for microbial, AD-related peptides [12] (Appendix A). Only a very small fraction of the screened peptides, 32 out of around 91 million peptides, returned significant hits (e-value < 1 × 10^−5^), but interestingly only against the APP. Most of the source proteins encrypting these APP-like peptides belonged to members of the order Clostridiales (85% of the peptides retrieving at least one significant match against the NCBI nonredundant database; e-value < 1 × 10^−5^), such as the genera *Clostridium* (15%), *Massilimaliae* (15%) or *Eubacterium* (10%), which are meaningful members of the human gut microbiome (Appendix A). 

Additionally, it is worth highlighting that 20 of these 32 peptides presented homology in the APP region corresponding to the Aβ42 peptide (Appendix A). Secondary and tertiary structure alignment of these bacterial peptides against the Aβ42 peptide revealed 13 of these 20 peptides had an Aβ42-like similitude score greater than 50% (Table 1 and Appendix A). This Aβ42-like score considers sequence, secondary and tertiary alignment, but adds weight to the secondary structure (2-fold compared with the others), since it has a greater influence on peptide bioactivity. All of these 13 peptides conserved, at least, the N-terminal alpha helix, and 2 of them even a second alpha helix, being thus more similar to the tertiary structure of the Aβ42 peptide (Figure 1A).

We selected the two best candidate peptides based on their Aβ42-like similitude score (>62%), namely AG25 and DF56 (Table 1). Both peptides were similar in secondary and tertiary structure to the Aβ42 peptide, were highly hydrophobic in terms of presence of aliphatic amino acids, and were therefore very likely to undergo the aggregative process that determines the toxicity and pathogenicity of Aβ42 peptide. The rationale behind the selection of these two bacteria-derived peptides was the possibility to deploy the biological effect of the Aβ42 peptide. As a preliminary in vitro model, we used neuronal, differentiated SH-SY5Y cells, which were cultivated in fresh EMEM/F12 medium supplemented with AG25 or DF56 peptides, tested at 1 and 2 ug/mL concentrations and with the corresponding DMSO controls (0.05% and 0.1% *v/v*, respectively). DMSO was used as the dilutant because, as pointed before, these peptides turned out to be highly hydrophobic. Considering the gene expression profiles, all the different conditions highly, but not completely, overlapped based on ordination (Figure 1B). However, the gene-set enrichment analysis results (Appendix A) showed that, while in most of the compared conditions no statistical differences were found, 83 pathways related to AD in the literature were found to be differentially expressed (*p*-value < 0.05, FDR < 0.2; Figure 1D; Appendix A) when comparing the DF56 2 ug/mL against the DMSO 0.1% condition. From them, seven pathways were already upregulated in the DMSO with respect to the control (EMEM/F12 medium) condition, and therefore were discarded for further analyses.

## 4. Discussion

First of all, it should be highlighted that this work provides preliminary and speculative evidence of novel mechanisms involved in AD, supported by the identification of a limited number of Aβ-like peptides encoded in the human gut microbiome. Our findings deserve therefore further mechanistic and translational research to better address the concept of gut microbiota-derived peptides contributing to AD. Although independent scientific works have established connections between gut microbiota and several neurological disorders, few studies have been focused in AD. Some interesting elements drew our attention to a potential relationship between our gut microbial populations and AD. Firstly, the fact that specific *Escherichia coli* strains are able to produce amyloid-like fibrils [14]. Secondly, Aβ peptide plaques have been observed in enteric neurons of mouse models of AD and in the submucosa of two AD patients [15]. Additionally, APPPS1-Tg mice, a mouse model of AD, showed reduced cerebral/serum Aβ peptide levels when bred under germ-free conditions [16].

Aβ peptides ranging from 38 to 43 amino acids are highly hydrophobic, Aβ42 being the most, which correlates with the highest amyloidogenic potential [4]. The two peptides selected in this work, AG25 and DF56, were also shown to be highly hydrophobic. For this reason, DMSO and nonpolar solutions were needed to dissolve them. One of the limitations of our mechanistic approach is that is unable to determine whether molecules such as AG25 and DF56, or other Aβ-like peptides derived from the gut microbiota, can cross the blood–brain barrier. It was shown that peptide DF56 triggered AD-related mechanisms in the neuron cell-line model. It is known that the Aβ peptide is present in peripheral blood, and recent results from Lam and colleagues, using genetically modified mice that produce human Aβ peptide in their livers, pointed to the fact that the peripheral metabolism of this peptide and APP might be related to AD risk [17]. These results strongly suggest that Aβ peptides and Aβ-like peptides might cross the blood–brain barrier.

Our knowledge about the relationship between the gut microbiota and AD is limited, and there is few and sparse data about gut microbiota dysbiosis in the context of AD. Here, we present a methodology for the identification of Aβ-like peptides encrypted in proteins of the human gut microbiota; our in vitro experiments, with two of these peptides challenging the SH-SY5Y neuronal cell line, represent the first molecular evidence of the implication on one bacterial derived peptide in AD. Peptide DF56 induced whole sets of genes representing pathways related to AD (the full list of references and pathways is shown in Appendix A). Over-expression of genes, such as *IRAK-1*, *NOTCH3*, *TNF receptor-1*, *STILTs* and *ROBOs* (Figure 1C), has been previously shown to be related to AD. A similar scenario was reported regarding the upregulation of pathways related to dysregulation of the mRNA translation, mitochondrial dysfunction, aggrephagy, fibrinolysis, and neuronal death, as well as metabolic pathways such as the purine and selenoamino acid metabolism. Moreover, pathways related to DNA repair, or to neuroprotective processes such as SUMOylation, have been found to be downregulated in the presence of the DF56 peptide.

## 5. Conclusions

To sum up, this exploratory work provides preliminary data on the contribution of peptides derived from the human gut microbiota to AD. One out of the two selected peptides was shown to affect a number of AD-related pathways, but in vitro testing of Aβ42-like peptides derived from gut bacteria needs to be increased. This study opens the way to new approaches for understanding the relationship between the human gut microbiota and the triggering of potential molecular events in the gut leading to AD. Given the limitations of the mechanistic data presented in our work, further research including the use of tools for targeted microbiota modification [18], microbiota transplant in relevant animal models or nutritional interventions will help us understand the individual roles of gut bacteria producing proteins encrypting Aβ-like peptides, and whether these Aβ-like peptides might promote, initiate or participate in the progress of AD.

## Figures and Tables

**Figure 1 nutrients-13-03868-f001:**
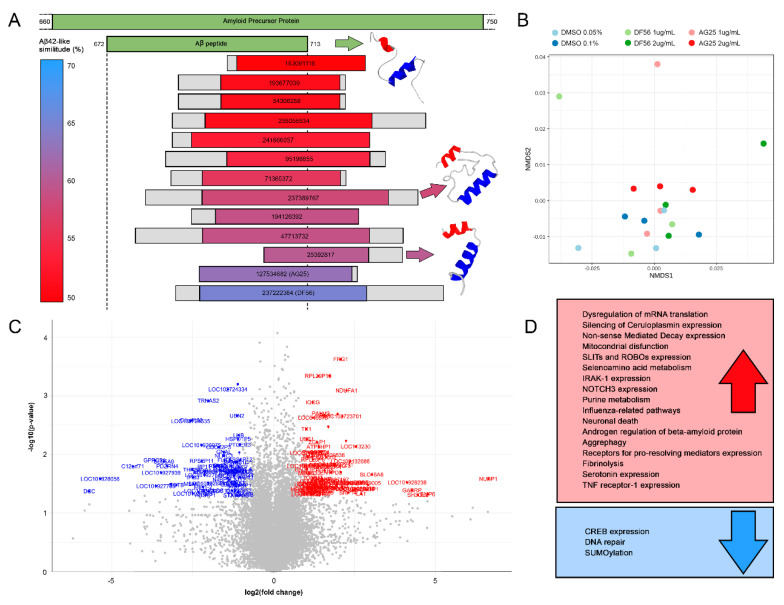
(**A**) From top to bottom, graphical representation of an alignment between human APP, Aβ42 and the 13 bacterial peptides contained in the MAHMI database that obtained the best Aβ42-like similitude score. The Aβ42-like similitude score considers sequence, secondary and tertiary alignment scores and is represented using a similitude gradient scale from red (50) to light blue (70). The coloured fraction of each peptide represents the sequence covered in the sequence alignment. Tertiary structure predictions for the peptides presenting two alpha helices are presented on the right, and are compared with the corresponding and resolved structure of the Aβ42 peptide obtained at the Protein Data Bank (https://www.rcsb.org/). (**B**) Nonmetric multidimensional scaling analysis based on the gene expression profiles of the different tested conditions. Bray–Curtis dissimilarity distances over log transformed FPKM (fragments per kilobase of exon model per million reads mapped) were used. (**C**) Volcano plot including genes influenced by DF56 peptide. Log2 of the fold changes (DF56 2 ug/mL compared to DMSO 0.1% *v/v*) is represented against the –log10 of the test *p*-values. Genes showing higher changes [|Log2(FC)| > 1 and –Log10(*p*-value) > 1.3] are highlighted in red (upregulated) or blue (downregulated). (**D**) Up- (blue) and down- (red) regulated pathways in neuronal differentiated SH-SY5Y cells by DF56 peptide (2ug/mL) compared to DMSO (0.1% *v/v*) control conditions. Only statistically differentially expressed pathways are reported (*p*-value < 0.05, FDR < 0.2).

**Table 1 nutrients-13-03868-t001:** Bacterial peptides contained in the MAHMI database that obtained the best score against the Aβ42 peptide. Aβ42-like similitude score considers sequence, secondary figure and tertiary alignment scores. BLAST = Basic Local Alignment Search Tool, eValue = expected value, MAX = maximum identity, AVG = average identity.

MAHMI Peptide	BLAST			Secondary Structure	Tertiary Structure		Aβ42-Like Similitude
	Identity	eValue	Coverage		MAX	AVG	
163091118	51.85%	3 × 10^−6^	33.33%	75.00%	54.03%	36.01%	50.82%
193677039	48.00%	3 × 10^−6^	40.48%	75.68%	38.00%	34.12%	51.22%
54306258	48.00%	3 × 10^−6^	40.48%	75.68%	39.13%	34.44%	51.31%
255055534	37.14%	6 × 10^−6^	50.00%	72.83%	43.09%	41.57%	51.45%
241666057	41.18%	7 × 10^−6^	54.76%	82.05%	33.51%	25.95%	53.15%
95198855	50.00%	5 × 10^−7^	38.10%	79.07%	44.91%	42.43%	54.90%
71365372	51.72%	2 × 10^−6^	50.00%	86.84%	32.89%	25.98%	56.38%
237389767	35.90%	7 × 10^−6^	50.00%	79.17%	61.26%	55.01%	57.82%
194126392	40.00%	4 × 10^−6^	42.86%	87.84%	48.28%	43.16%	59.00%
47713732	42.86%	4 × 10^−6^	50.00%	91.67%	36.32%	33.23%	59.50%
25392817	54.55%	5 × 10^−6^	19.05%	89.71%	57.46%	51.02%	60.21%
127534682 (AG25)	43.75%	3 × 10^−6^	52.38%	88.89%	54.08%	48.46%	62.29%
237222364 (DF56)	45.71%	1 × 10^−6^	50.00%	92.71%	54.42%	48.58%	64.21%

## Data Availability

The data that support the findings of this study are available from the corresponding author, B.S., upon reasonable request.

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
