# Peer review of "Microbiota-Derived β-Amyloid-like Peptides Trigger Alzheimer’s Disease-Related Pathways in the SH-SY5Y Neural Cell Line"

_nutrients, 2021, doi:10.3390/nu13113868_

Round 1

Reviewer 1 Report

This “Communication” report by Blanco-Miguez et al provides preliminary evidence that peptides produced by the gut microbiota may contribute to Alzheimer’s disease due to their similarity with amyloid-β (Aβ) peptides. The concept of gut-derived peptides contributing to AD is interesting; however, the present data provide only preliminary and speculative evidence considering the small number of Aβ-like peptides identified and limited mechanistic data included. Nevertheless, the data provide an interesting provide a novel framework for identifying novel mechanisms of AD. 

Major Comments: 

  • The results of this study are very preliminary and should be described as such throughout the paper. The authors should consider describing this as an “exploratory” analysis to provide “proof-of-concept" evidence that gut-derived peptides may activate AD-related mechanisms. 
  • 32 out of ~91M peptides tested for AD-related peptides in the MAHMI database seems exceedingly small. The authors should further explain why these peptides are considered important and not likely to be statistical noise. Including the e-value is helpful, but its meaning should be further explained for readers without a background in bioinformatics. 
  • The rationale for only testing two of the 13 peptides included in Figure 1A in cell culture experiments is not clearly explained or justified. Inclusion of additional peptides may have provided stronger evidence in support of the overall hypothesis that these peptides activate AD-like mechanisms. 
  • Justification for using 1 and 2 µg/ml of each peptide in cell culture experiments should also be included. Are Aβ-like peptides from the gut likely to reach the neurons of the brain at these concentrations? 
  • The limitations of the study should be included in the discussion, including some discussion about whether gut-derived Aβ-like peptides can cross the blood brain barrier, especially considering the statement on line 46 that Aβ42 is more hydrophobic than other peptides. 
  • The authors may consider citing recent work by Lam et al (PMID: 34520451) as further evidence that Aβ originating outside of the brain (in this case the liver) can contribute to AD-like pathology. 
  • Some graphical representation (e.g., fold-change) of the top genes influenced by DF56 and AG25 in cell culture experiments should be included in the primary results section rather than just the list of ontology terms. 

Minor Comments: 

  • Figure 1A: Similitude scale should include “%” as the unit to be more clear 
  • Abstract: Suggest changing “neuropeptide peptides” to “neuropeptides” in first sentence. 
  • Line 65-66: I think the word “suggested” should be changed to “subjected” 
  • Line 102: “hydrophobics” should be changes to “hydrophobic” 

Author Response

Manuscript ID nutrients-1360427

Manuscript´s Title: Presence of β-amyloid-like peptides encrypted in the human gut microbiome. A new target for dietary interventions in Alzheimer’s Disease.

Responses to Reviewer 1

Comments to the author:

This “Communication” report by Blanco-Miguez et al provides preliminary evidence that peptides produced by the gut microbiota may contribute to Alzheimer’s disease due to their similarity with amyloid-β (Aβ) peptides. The concept of gut-derived peptides contributing to AD is interesting; however, the present data provide only preliminary and speculative evidence considering the small number of Aβ-like peptides identified and limited mechanistic data included. Nevertheless, the data provide an interesting provide a novel framework for identifying novel mechanisms of AD.

Reviewer’s major comments:

Major comment 1:

The results of this study are very preliminary and should be described as such throughout the paper. The authors should consider describing this as an “exploratory” analysis to provide “proof-of-concept" evidence that gut-derived peptides may activate AD-related mechanisms.

Author's response:

We understand the concern raised by the reviewer and apologise for the unintended lack of clarity. In fact, we have chosen this journal as it allows publication of original scientific works presenting preliminary but sounding research results, under the formula of short communication. To better illustrate this point, we have included several references to the exploratory character of this study, notably in the abstract, end of the introduction section and in the result/discussion/conclusion. We have reviewed the manuscript to better reflect the preliminary of our findings.

Major comment 2:

32 out of ~91M peptides tested for AD-related peptides in the MAHMI database seems exceedingly small. The authors should further explain why these peptides are considered important and not likely to be statistical noise. Including the e-value is helpful, but its meaning should be further explained for readers without a background in bioinformatics.

Author's response:

We understand the point raised by the reviewer; further than the statistical analysis the potential biological significance of these peptides has been better explained, further than bioinformatics reasons, in the results section (Line 159).

Major comment 3:

The rationale for only testing two of the 13 peptides included in Figure 1A in cell culture experiments is not clearly explained or justified. Inclusion of additional peptides may have provided stronger evidence in support of the overall hypothesis that these peptides activate AD-like mechanisms.

Author's response:

We understand the criticism raised by the referee in terms of the low number of peptides tested, but this work, as explained and clarified now through the MS, is a proof-of-concept. These two peptides were selected following not only global bioinformatics criteria, but also because the predicted secondary and tertiary structures were somehow similar to those of the Aβ42. Indeed, one out of the two peptides had no effect over cell line transcriptomes at all. We have included a sentence at the end of the MS stating that number of Aβ42-like peptides, derived from gut bacteria, needs to be increased.

Major comment 4:

Justification for using 1 and 2 µg/ml of each peptide in cell culture experiments should also be included. Are Aβ-like peptides from the gut likely to reach the neurons of the brain at these concentrations?

Author's response:

We have included a new Supplementary Figure in which selection of peptide concentrations is conveniently expanded and explained. Briefly, a RTCA-DP (Real Time Cell Analyzer) xCELLigence (from ACEA Biosciences Inc., Agilent Technologies) was used to select the most suitable concentrations to test the effect of both peptides over the neuron cell line model.

Major comment 5:

The limitations of the study should be included in the discussion, including some discussion about whether gut-derived Aβ-like peptides can cross the blood brain barrier, especially considering the statement on line 46 that Aβ42 is more hydrophobic than other peptides.

Author's response:

The two peptides selected in this work are also highly hydrophobic. They were not soluble in water-based solutions, and DMSO was used as solvent. We have included a paragraph in the discussion stating this point, pointing also to the limitations of our mechanistic approach. Abstract has been also modulated to reflect that further research is needed to show the potential relationship of these gut microbiota derived peptides with AD.

Major comment 6:

The authors may consider citing recent work by Lam et al (PMID: 34520451) as further evidence that Aβ originating outside of the brain (in this case the liver) can contribute to AD-like pathology.

Author's response:

We thank the reviewer for the suggestion. We have included this work in the discussion, integrated in the response to Major comment #5.

Major comment 7:

Some graphical representation (e.g., fold-change) of the top genes influenced by DF56 and AG25 in cell culture experiments should be included in the primary results section rather than just the list of ontology terms.

Author's response:

Following the reviewer’s suggestion, we have included a volcano plot of the top genes influenced by the presence of both peptides. Please, see the new version of Figure 1.

Reviewer’s minor comments:

Figure 1A: Similitude scale should include “%” as the unit to be more clear

Abstract: Suggest changing “neuropeptide peptides” to “neuropeptides” in first sentence.

Line 65-66: I think the word “suggested” should be changed to “subjected”

Line 102: “hydrophobics” should be changes to “hydrophobic”

Author’s response:

All minor concerns have been fixed following the reviewer’s recommendations.

Reviewer 2 Report

In this study authors tried to show in silico and in vitro samples of Aβ-like peptides released from the gut microbiome. They tested two peptides with homology to human Aβ peptide domain in a neuron cell line model.

The authors described that Gene expression profile analysis showed that one of these peptides induced whole gene pathways related to AD. The authors finally concluded that these peptides might be implicated in the neurodegenerative process leads to AD.

Major comments:

The topic of the gut microbiome and its role In AD is interesting. Anyhow here authors used samples from database and with just in vitro study from a data base without in vivo works as well as human studies ( i.e autopsy of AD patients) it is very difficult to conclude as authors did.  For example In title:  Presence of β-amyloid-like peptides encrypted in the human gut microbiome. A new target for dietary interventions in Alzheimer’s Disease. I believe this does not show a target for dietary intervention because there is no animal or human direct study to prove this.

How do the authors prove that these Aβ-like peptides that they found in this database can pass the blood brain barrier? How it may have role to promote the AD without knowing its effect in vivo and human? I believe it is not easy to conclude as authors did in line 25: these peptides might be implicated in the neurodegenerative process that initiates, promotes or mediates AD.

I would suggest changing the title to be more specific title related to this study. In addition, I would recommend to the authors to be more precise in concluding parts as this study is in vitro and cannot prove and answer the main questions. As a result, I would suggest to the authors to revise the text.

Minor comments:

The abstract started with : Previously, we have reported on the presence of potential neuropeptide peptides in human gut microbiota using the MAHMI database. Surprisingly, few of them  showed similarity to β-amyloid peptide (Aβ). Independent scientific works have shown  relationships between the gut microbiome and several neurological disorders. Although  few studies have been focused on Alzheimer’s Disease (AD), it has been already shown  the ability of some bacteria to produce amyloid-like fibrils and the accumulation of β amyloid (Aβ) peptide plaques in enteric neurons of mouse models of AD and in the submucosa of AD patients.

I would suggest to the authors to move this part to the introduction.

 The text is not written according to the Nutrients journal guidelines. For examples, the references both in the text and reference list should be numerical.

Author Response

Manuscript ID nutrients-1360427

Manuscript´s Title: Presence of β-amyloid-like peptides encrypted in the human gut microbiome. A new target for dietary interventions in Alzheimer’s Disease.

Responses to Reviewer 2

Comments to the author:

In this study authors tried to show in silico and in vitro samples of Aβ-like peptides released from the gut microbiome. They tested two peptides with homology to human Aβ peptide domain in a neuron cell line model.

The authors described that Gene expression profile analysis showed that one of these peptides induced whole gene pathways related to AD. The authors finally concluded that these peptides might be implicated in the neurodegenerative process leads to AD.

Reviewer’s major comments:

Major comment 1:

The topic of the gut microbiome and its role in AD is interesting. Anyhow here authors used samples from database and with just in vitro study from a data base without in vivo works as well as human studies (i.e autopsy of AD patients) it is very difficult to conclude as authors did.  For example, in title:  Presence of β-amyloid-like peptides encrypted in the human gut microbiome. A new target for dietary interventions in Alzheimer’s Disease. I believe this does not show a target for dietary intervention because there is no animal or human direct study to prove this

Author's response:

Following the reviewer’s suggestion, and in general the concerns raised by the two referees of these MS in the sense of the exploratory character of our work and the limitations of our mechanistic insights, the whole MS has been reviewed to better reflect the preliminary of our findings.

Major comment 2:

How do the authors prove that these Aβ-like peptides that they found in this database can pass the blood brain barrier? How it may have role to promote the AD without knowing its effect in vivo and human? I believe it is not easy to conclude as authors did in line 25: these peptides might be implicated in the neurodegenerative process that initiates, promotes or mediates AD.

Author's response: As replied to reviewer 1, the two peptides selected in this work were also highly hydrophobic. They were not soluble in water-based solutions, and DMSO was used as solvent. We have included a paragraph in the discussion stating this point, pointing also to the limitations of our mechanistic approach. Further, we have also included a recent reference from Lam and colleagues that pointed to a potential relationship between peripheral metabolism of AB peptides with AD risk. We have also modulated the sentence in line 25 to better reflect our mechanistic limitations.

Major comment 3:

I would suggest changing the title to be more specific title related to this study. In addition, I would recommend to the authors to be more precise in concluding parts as this study is in vitro and cannot prove and answer the main questions. As a result, I would suggest to the authors to revise the text.

Author’s response:

We agree with the reviewer. Our suggestion for a more specific title is “Microbiota derived β-amyloid-like peptides trigger Alzheimer’s Disease-related pathways in the SH-SY5Y neural cell line”. Further, we have reviewed the conclusions section (and in general the whole MS) to better reflect the preliminary of our findings.

Reviewer’s minor comments:

Minor comment 1:

The abstract started with : Previously, we have reported on the presence of potential neuropeptide peptides in human gut microbiota using the MAHMI database. Surprisingly, few of them  showed similarity to β-amyloid peptide (Aβ). Independent scientific works have shown  relationships between the gut microbiome and several neurological disorders. Although  few studies have been focused on Alzheimer’s Disease (AD), it has been already shown  the ability of some bacteria to produce amyloid-like fibrils and the accumulation of β amyloid (Aβ) peptide plaques in enteric neurons of mouse models of AD and in the submucosa of AD patients.

I would suggest to the authors to move this part to the introduction.

Author’s response:

Following the reviewer’s suggestion, we have moved and integrated (to avoid duplicates) this section to the introduction.

Minor comment 2:

The text is not written according to the Nutrients journal guidelines. For examples, the references both in the text and reference list should be numerical.

Author’s response:

We apologize for the inadvertently formatting error. The reference style has been reviewed according to the journal guidelines.

Round 2

Reviewer 2 Report

The authors addressed my concerns.